# Cultivation of *Solanum lycopersicum* under Glass Coated with Nanosized Upconversion Luminophore

**Dmitry E. Burmistrov** [1], **Denis V. Yanykin** [1,2,*], **Alexander V. Simakin** [1], **Mark O. Paskhin** [1], **Veronika V. Ivanyuk** [1], **Sergey V. Kuznetsov** [1], **Julia A. Ermakova** [1], **Alexander A. Alexandrov** [1] and **Sergey V. Gudkov** [1]

1 Prokhorov General Physics Institute of the Russian Academy of Sciences, 38 Vavilova St., 119991 Moscow, Russia; dmitriiburmistroff@gmail.com (D.E.B.); avsimakin@gmail.com (A.V.S.); pashin.mark@mail.ru (M.O.P.); veronika.ivaniuk@yandex.ru (V.V.I.); kouznetzovsv@gmail.com (S.V.K.); julia.r89@mail.ru (J.A.E.); alexandrov1996@yandex.ru (A.A.A.); S_makariy@rambler.ru (S.V.G.)
2 Institute of Basic Biological Problems, Russian Academy of Sciences, 2 Institutskaya St., Pushchino, 142290 Moscow, Russia
* Correspondence: ya-d-ozh@rambler.ru

**Abstract:** The effect of upconverting luminescent nanoparticles coated on glass on the productivity of *Solanum lycopersicum* was studied. The cultivation of tomatoes under photoconversion glass led to an increase in plant productivity and an acceleration of plant adaptation to ultraviolet radiation. An increase in the total leaf area and chlorophyll content in the leaves was revealed in plants growing under the photoconversion glass. Plants growing under the photoconversion glass were able to more effectively utilize the absorbed light energy. The results of this study suggest that the spectral changes induced by photoconversion glass can accelerate the adaptation of plants to the appearance of ultraviolet radiation.

**Keywords:** photoconversion; upconversion; nanomaterials; greenhouses; insufficient insolation; *Solanum lycopersicum*

## 1. Introduction

Luminescent materials are used in the development of optical devices and biomedical equipment for water splitting, $CO_2$ reduction, environmental purification, photosynthesis and agriculture. At present, different luminescent materials have been developed, such as metal complexes, semiconductors, nanoparticles, fluorescent proteins, organic dyes and luminophores based on donor–acceptor pairs within a metal–organic framework [1–5]. Luminophores are interesting in that they serve as a binder for nanoparticles and can themselves be an additional photoconverter of sunlight [5]. Of specific interest is the use of luminophores based on upconversion nanoparticles. Upconversion luminescence is a process in which low-energy light is converted to higher-energy light, through multiple photon absorptions or energy transfers from sensitizers (used to absorb light energy) to activator ions. Light upconversion was theoretically predicted in 1931 by Goeppert Mayer [6]. Researchers have noted several groups of mechanisms through which the upconversion processes occur. The first is energy transfer upconversion; the second is excited-state absorption; and the third is a process named the photon avalanche [7–11]. Upconversion luminophores do not seem very promising for biological and medical applications [12–14], for the preparation of solar cell concentrators [15] or as photoconversion coatings for the transparent shells of greenhouses. However, in comparison with other emission processes based on multiphoton absorption, upconversion can be efficiently excited even at low densities of excitation illuminations [15]. The distance, mutual orientation and concentration of doped ions are especially important for the realization of highly efficient energy transfer upconversion [16]. The most efficient upconversion is present in solid-state materials doped with rare-earth ions. At present, high-quality upconversion nanocrystals

can be synthesized, with particle size, shape and optical properties being controlled. The development of upconversion luminophores that convert near-infrared radiation (IR) into visible light started in the middle of the XXth century [17]. At present, it is considered promising to use combinations of rare-earth elements (REE) (for example, $Yb^{3+}$: $Er^{3+}$, $Yb^{3+}$: $Tm^{3+}$ and $Yb^{3+}$: $Ho^{3+}$ ions) to create visible light sources. $Yb^{3+}$ is used as a sensitizer due to its simple energy levels, structure and large absorption cross-section [18], while $Er^{3+}$, $Tm^{3+}$ and $Ho^{3+}$ ions are used as activators [7,17,19]. Strontium fluoride is usually used as a matrix, which provides a high quantum yield of upconversion [20,21]. At present, the maximum quantum yields reported for upconversion luminophores in the solid state are about 9–12% [22,23]. Despite the fact that the idea of creating photoconversion films appeared more than thirty years ago [24], upconversion luminophores do not apply to the preparation of photoconversion coatings for the transparent shells of greenhouses due to small spectral changes. However, in many studies [25–46], despite an insignificant increase in light intensity in the PAR region, a positive effect of photoconversion coatings on plant growth was shown. Previously, it was shown that moderate concentrations of rare-earth-based upconversion nanoparticles did not inhibit the growth of mung beans, and these plants did not become toxic to mice [47]. The authors concluded that nanoparticles based on rare-earth elements can be widely used due to their non-toxicity to plants and animals. In this work, an investigation of the effect of upconverting luminescent nanoparticles incorporated into fluoropolymer films coated on glass on the growth of agricultural plants was performed. $Sr_{0.910}Yb_{0.075}Er_{0.015}F_{2.090}$ nanoparticles were used as the upconverting luminophore, which converts IR into visible light.

## 2. Materials and Methods

Preparation of Photoluminophore Nanoparticles and Investigation of their Properties: The synthesis of upconversion nanopowders with the nominal composition $Sr_{0.910}Yb_{0.075}$ $Er_{0.015}F_{2.090}$ was carried out by co-precipitation from aqueous nitrate solutions under ambient conditions, followed by low-temperature drying at 45 °C and high-temperature annealing at 600 °C, according to a previously published protocol [48].

To manufacture the photoconversion composites, a 7% solution of nanoparticles in acetone mixed with the liquid component of the fluoroplate polymer was used [26]. A photoconversion film (PCF) containing luminescent nanoparticles was formed on the glass surface using a spray gun.

X-ray pattern diffraction (XRD) was performed on a BRUKER D8 ADVANCE diffractometer (Billerica, Massachusetts, USA) with $CuK\alpha$ radiation. The unit cell parameters (a) were calculated in a fluorite structure (Fm-3m space group) using POWDER 2.0 software. The morphology and particle size of the nanoparticles were determined using a Carl Zeiss NVision 40 microscope (Zeiss AG, Oberkochen, Germany) connected with an Oxford Instruments XMAX (80 mm$^2$) set-up (Oxford Instruments *plc*, Abingdon, UK) (using ImageJ software). The hydrodynamic diameter and zeta potential distribution of the nanoparticles were measured with a Zetasizer Ultra (Malvern Panalytical, Malvern, UK). The upconversion luminescence spectra in the visible range were recorded using a fiber-optic spectrometer USB2000 (OceanOptics, Orlando, Florida, USA). The sample was placed inside the integrating sphere Cintra 4040 (GBC Scientific, Braeside, Victoria, Australia) and irradiated using a 50 mW IR light diode, which emits in the region of 960–985 nm with a maximum at 975 nm. The upconversion emission and the scattered laser radiation were collected by the fiber and delivered to the spectrometer.

Plant Material and Growth Conditions: *Solanum lycopersicum* plants were used in the study. The plants were grown from seeds. Before the experiments, all plants were grown in a 16 h light/8 h dark cycle at 25–26 °C and at a weak light intensity (PPFD (400 nm–700 nm) and PFD (350 nm–800 nm) were 70 µmol photon s$^{-1}$ m$^{-2}$ and 140 µmol photon s$^{-1}$ m$^{-2}$, respectively). At the seventh leaf stage, both experiment and control plants were placed under glass coated with a fluoroplate polymer with or without photoconversion nanoparti-

cles. A UVA component ($\lambda$ = 370 nm, PFD = 10 µmol photon s$^{-1}$ m$^{-2}$) was added to the illumination spectrum.

The leaf area was determined using the GreenImage software developed by our team [26]. The content of chlorophyll in the leaves was determined using a CL-01 chlorophyll content meter (Hansatech Instruments, Norfolk, UK). The number of leaves and the stem length were determined manually. The light intensity was measured using a PG200N Spectral PAR Meter (UPRtek, Zhunan, Miaoli, Taiwan).

Chlorophyll Fluorescence (ChlF) Measurement: Before the experiments (treatments and measurements), both tomato plants and leaves were kept for 60 min in the dark at a temperature of 25–26 °C. The kinetics of photoinduced changes in the ChlF yield ($\Delta$F) related to the photoreduction of the primary electron acceptor, $Q_A$, were measured on a non-cut leaf at room temperature with a DUAL-PAM-100 fluorometer (Walz, Eichenring, Effeltrich, Germany). To measure the maximum quantum yield of PS2 photochemistry ($(Fm - F_0)/Fm$, where $F_0$ is the initial fluorescence value and Fm is the maximum fluorescence value) and ChlF parameters after light adaptation (10 min, $\lambda$ = 625 nm, 250 µmol photon s$^{-1}$ m$^{-2}$), the samples were illuminated with a saturating 500 ms flash ($\lambda$ = 625 nm, 12,000 µmol photon s$^{-1}$ m$^{-2}$). The calculation of the ChlF parameters was performed using DualPAM software [49].

Statistical Analysis: To determine any statistically significant differences between the plant groups, a one-way analysis of variance (ANOVA) followed by post hoc comparisons using Tukey's test and Student's *t*-test for independent means was performed. The difference was considered significant if $p \leq 0.05$.

## 3. Results

It was previously shown that illumination of $SrF_2$ doped with $Er^{3+}$ and $Yb^{3+}$ single crystals with infrared light induces luminescence in red and green spectral regions [21,23,50–53]. Similar spectra in these regions were obtained in the present work (Figure 1A, curve 1). Infrared (975 nm) excitation of $Sr_{0.910}Yb_{0.075}Er_{0.015}F_{2.090}$ (UCL) led to photoluminescence with typical emission bands of the ion pair $Er^{3+}/Yb^{3+}$: 1.87 eV (about 660 nm), 2.27 eV (545 nm) and 2.36 eV (525 nm). Thus, the obtained results allow us to assume that the selected luminescence nanoparticles have sufficient potential to improve light for plant growth and may be used in photoconversion covers to increase the PAR intensity due to IR conversion into red and green light.

Figure 1B shows a typical X-ray pattern diffraction for a $Sr_{0.910}Yb_{0.075}Er_{0.015}F_{2.090}$ solid solution dried at 45 °C in air (curve 1) and treated at 600 °C in a platinum crucible (curve 2). The synthesis of the solid solution $Sr_{1-xy}Yb_xEr_yF_{2+ x+y}$ leads to the formation of a single-phase powder with a fluorite structure (JCPDS #06−0262, $a$ = 5.800 Å) (compare curve 1, curve 2 and curve 3). As it can be seen from the figure, the powder dried in air at 45 °C (curve 1) has broadened peaks that indicate a small particle size. Heat treatment at 600 °C led to narrowing of the peaks (curve 2) due to the agglomeration of particles. The unit cell parameters were $a$ = 5.7944(2) and $a$ = 5.7722(1) for the 45 °C and 600 °C thermal treatment stages, respectively. The scanning electron microscopy image of the samples shows that heat treatment led to the formation of rounded particles with a 68 and 350 nm mean size (Figure 1C), which is consistent with the XRD data. The dynamic light scattering method confirmed that the powder contained two pools of particles: primary particles with an average nanoparticle size from 55 to 85 nm, and agglomerates of primary particles (Figure 1D). Note that most of the particles are presented in the form of agglomerates. The real composition of the solid solution powder was determined using energy-dispersive X-ray spectroscopy on $Sr_{0.897}Yb_{0.088}Er_{0.015}F_{2.103}$.

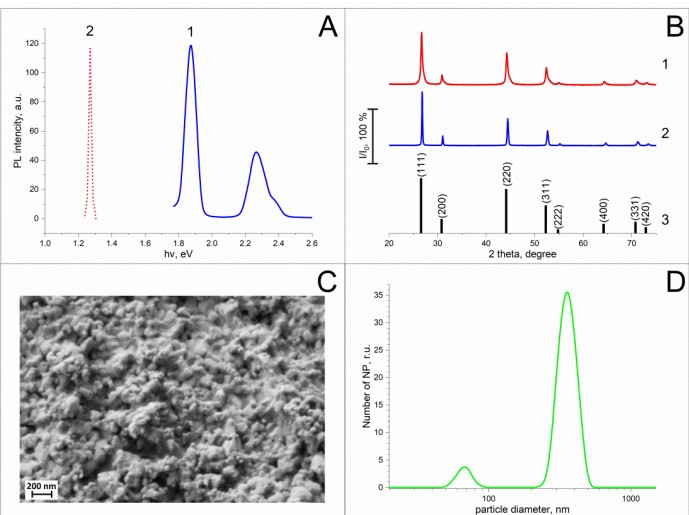

**Figure 1.** Characteristics of nanosized fluorophores ($Sr_{0.910}Yb_{0.075}Er_{0.015}F_{2.090}$). (**A**) Photoluminescence of the nanoparticles in acetone in the PAR range (1), which were excited with a semiconductor laser (2). It should be noted that the emission/excitation ratios presented in the figure do not correspond to real quantum yields of photoconversion. (**B**) X-ray diffraction pattern for the nanoparticles after heat treatment at 45 °C (1), 600 °C (2) and JCPDS #06-0262 for $SrF_2$ (3). (**C**) SEM images for samples after heat treatment at 600 °C. (**D**) Size distribution of the nanoparticles of fluorophores obtained using dynamic light scattering.

Our measurements show that the average solar lighting on a cloudy day in the Moscow region reached a PPFD (400–700 nm) intensity of approximately 70–80 µmol photon $s^{-1}$ $m^{-2}$. To simulate natural light conditions, all of the plants were grown under 70 µmol photon $s^{-1}$ $m^{-2}$ (provided by an incandescent light bulb). At the seventh leaf stage, both experiment and control plants were placed under glass coated with a fluoroplate polymer with or without photoconversion nanoparticles. A UVA component ($\lambda$ = 370 nm, PFD = 10 µmol photon $s^{-1}$ $m^{-2}$) was added to the illumination spectrum.

The spectral composition of light can influence plant morphology, physiology and development, including photosynthesis [54–56]. It is known that light energy absorbed by plants' photosynthetic antenna complexes and transformed into excitation energy can be utilized via three processes: (1) photosynthesis, (2) regulated heat dissipation and (3) non-regulated heat dissipation and fluorescence emission (NO). The partitioning of absorbed light energy was determined by the analysis of the light-induced changes in the ChlF. The efficiency of photosynthesis correlates well with the effective quantum yield of photosystem 2 photochemistry (Y(II)), or so-called photochemical quenching of fluorescence ($q_P$). Regulated heat dissipation is reflected in the non-photochemical quenching of fluorescence (NPQ). Figure 2A,B show that the proportion of absorbed light energy used for photosynthesis before the start of the experiment was about 30%. At the same time, the maximum quantum yield ($_\Delta F/F_m$) in the leaves of the dark-adapted plants was equal to 0.82, which may indicate the good physiological state of the plants. $_\Delta F/F_m$ remained practically unchanged throughout the experiment (Figure 2A,B curves). However, a small decrease in $_\Delta F/F_m$ was still observed in the first seven days in the control plants (Figure 2A), which may indicate some disturbance in the functioning of the photosynthetic apparatus.

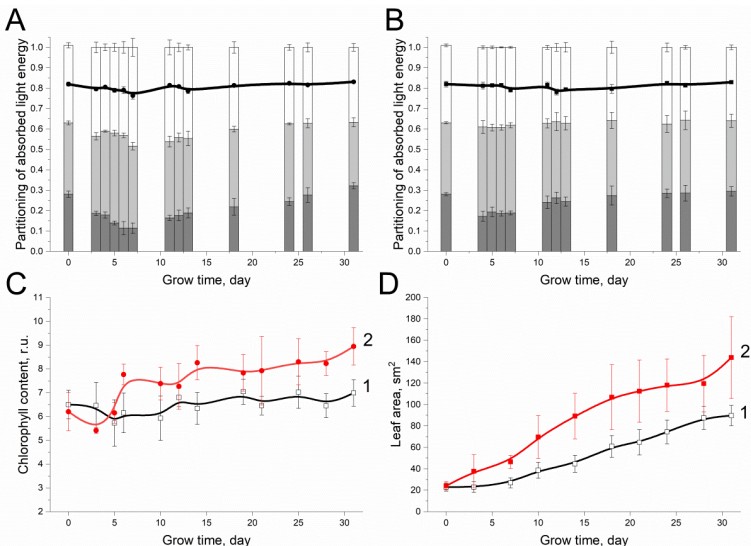

**Figure 2.** Dependence of the partitioning of absorbed light energy on the growth time of plants under glass coated with fluoroplate polymer with (**A**) or without (**B**) photoconversion nanoparticles. Effective quantum yield of PSII (Y(II)), quantum yield of light-induced non-photochemical quenching of ChlF (Y(NPQ)) and quantum yield of non-regulated heat dissipation and fluorescence emission (Y(NO)) are shown in dark gray, light gray and white, respectively. The black line indicates the maximal quantum yield of the photosystem II photochemistry. Dependence of relative chlorophyll content (**C**) and total area (**D**) of leaves on the growth time of plants under glass coated with fluoroplate polymer without (1) or with (2) photoconversion nanoparticles. The data are the means of at least 11 measurements, with the standard error of the mean. Relative chlorophyll content value is the mean of 4–10 measurements on at least 11 plants.

It was shown that the proportion of Y(II) decreased within the first days of the experiment and then was gradually restored to the initial values in the leaves of both plant groups. In the control plants, Y(II) drastically reduced within the first week from 0.28 to 0.11 (Figure 2B) and afterward increased slowly and reached the initial values by the end of the fourth week of the experiment. In contrast to the control plants, the tomatoes grown under the PCF demonstrated a relatively small decrease in Y(II) (from 0.28 to 0.19 within three–four experimental days), and a rapid restoration of Y(II) that reached the initial values by the end of the experiment (Figure 2A). A decrease in the proportion of Y(II) was accompanied by an increase in the heat dissipation of the excitation energy. In the control plants, Y(NO) increased by 25% and, within three weeks, relaxed to the values observed before the start of the experiment, while in the plants grown under the PCF, Y(NO) remained unchanged. At the same time, the development of Y(NPQ) in both plant groups had identical kinetics and amplitudes. Thus, differences in the development of the absorbed light energy redistribution in the control and experimental plant groups may reflect differences in both the inhibition of photosynthesis and activation of the heat dissipation of absorbed light energy during the first week of the experiment.

Differences in the efficiency of the photochemical reactions in the plants grown under common and photoconversion films caused a distinction in the plants' productivity (Figure 3). It was shown that the presence of photoconversion films above the plants had a moderate positive effect (without statistically significant differences) on the leaf formation rate and stem length. Nevertheless, the photoconversion films caused a gradual increase in the chlorophyll content in the leaves, while the chlorophyll content in the control group of plants remained practically unchanged during the experiment (Figure 2C). The effect of the PCF on the increase in the total leaf area was more significant: from about 23 to 90 sm$^2$ in the case of the control group and to 144 sm$^2$ in the case of tomatoes growing under the photoconversion film (Figure 2D). Thus, acceleration of plant adaptation to changes in the light conditions under the film containing photoconversion nanoparticles was shown.

This, in turn, led to an increase in the biomass of leaves and to a magnification in the accumulation of chlorophyll in them.

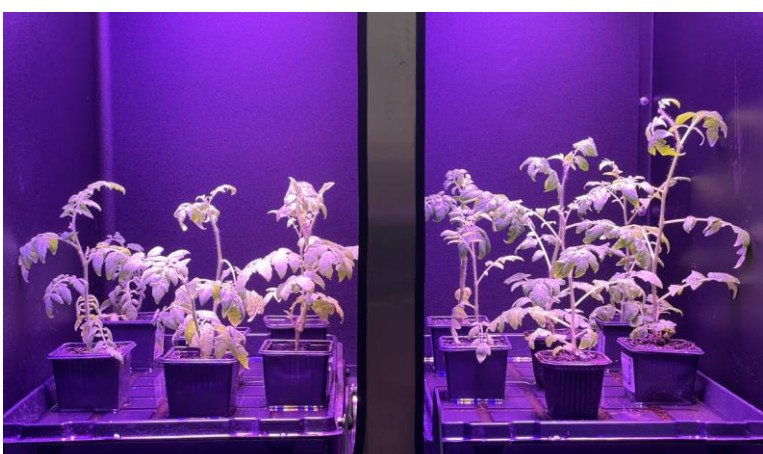

**Figure 3.** Photo of tomatoes grown under glass coated with a fluoroplate polymer without (**left**) or with (**right**) photoconversion nanoparticles.

## 4. Discussion

In our experiments, tomatoes adapted to growth under certain light conditions (low light intensity, absence of UV radiation and a high proportion of low-energy photons compared to photosynthetically active radiation) were exposed to a serious stress factor: a change in the spectral composition of light. On the one hand, the application of fluoroplate polymer coatings on the glass under which the studied plants grew led to a slight decrease in the light intensity. On the other hand, an ultraviolet component was applied to the spectrum, which can lead to some inhibition of plant growth and negatively affect their development. Figure 2D (curve 1) shows that changes in the light spectra led to a delay in plant growth in the first days of the experiment. After the first week, plant growth was restored. A similar pattern was not only observed in an increase in leaf area but also in the length of the stem and the number of leaves (data not shown). The growth of the plants under the photoconversion coating was not inhibited, even in the first days of the experiment (Figure 2D, curve 2), which caused the differences between the size of the plants in both groups. Moreover, in the leaves of the plants grown under the photoconversion film, an increase in the content of photosynthetic pigments (chlorophyll) was observed, indicating the development of adaptation processes. It is important to note that under the PCF, an increase in the chlorophyll content from $6.2 \pm 0.8$ to $8.9 \pm 0.1$ r.u. corresponds to an increase from 20–25 mg to approximately 40 mg Chl $(g\ FW)^{-1}$ [57], which may indicate significant rearrangements in the photosynthetic apparatus of the plants. Indeed, a change in the lighting conditions not only caused morphological changes in the plants but also induced transformations in the photosynthetic apparatus, reflected in changes in the kinetic parameters of chlorophyll a fluorescence. The inhibition of plant growth induced by changes in the light spectra was accompanied by both a decrease in Y(II) and an increase in the proportion of heat dissipation of the absorbed light energy. Moreover, this regularity was more pronounced in the control group of plants, in which heat dissipation was caused by both regulated and non-regulated heat dissipation, while in plants growing under the photoconversion film, only Y(NPQ) was observed. The fast restoration of the initial values of the effective quantum yield of photosystem II photochemical reactions in plants growing under the PCF may reflect the activation of a mechanism for eliminating the imbalance between photosynthetic electron transport and the utilization of NADPH. At the same time, the increase in Y(NO) that was only observed in the control plant group may indicate some damage to the photosynthetic apparatus.

Before making assumptions about the reasons for the positive effect of the PCF, it should be noted that the passing of light through glass coated with luminescent nanopar-



ticles was not accompanied by a noticeable increase in the intensity of the visible range. However, in the present work, as well as in a number of other works, an increase in plant productivity was demonstrated [25–31,33–46]. Nanosized particles can act as a protectant against UV owing to their capability to absorb and scatter ultraviolet radiation. Moreover, $SrF_2:Yb^{3+}(x)$- and $Er^{3+}(y)$-type nanoparticles can take part in the conversion of UV to red light. It was shown that the UV excitation (375 nm) of the $SrF_2:Yb^{3+}(x)$ and $Er^{3+}(y)$ nanoparticles with various $Yb^{3+}/Er^{3+}$ ratios led to an emission in the PAR region [21]. Conversion of IR to red light can change the red/far red ratio and thus activate the phytochrome system. Activation of the phytochrome system, in turn, could intensify photosynthesis, increase stress resistance and accelerate plant growth [29,58,59]. Previously, it was shown that photoconversion films stimulated an increase in the concentration of native soil microflora, which has a positive effect on plant growth [36,60]. The notion that, in light deficiency conditions, minor alterations in the light spectrum induced by photoconversion nanoparticles can accelerate the adaptation of plants to UV radiation is also not excluded.

Thus, it can be suggested that the spectral changes induced by photoconversion films coated on glass can accelerate the acclimation of plants when the weather changes and activate plant growth.

**Author Contributions:** Conceptualization, D.V.Y. and S.V.G.; methodology, D.V.Y., S.V.K. and S.V.G.; formal analysis, D.V.Y. and D.E.B.; investigation, D.E.B., D.V.Y., A.V.S., M.O.P., V.V.I., J.A.E., S.V.K. and A.A.A.; writing—original draft preparation, D.V.Y.; writing—review and editing, D.V.Y., S.V.K. and S.V.G.; funding acquisition, S.V.G. All authors have read and agreed to the published version of the manuscript.

**Funding:** This research was funded by a grant from the Ministry of Science and Higher Education of the Russian Federation for large scientific projects in priority areas of scientific and technological development (subsidy identifier: 075-15-2020-774).

**Institutional Review Board Statement:** Not applicable.

**Informed Consent Statement:** Not applicable.

**Acknowledgments:** The authors are thankful to V.V. Voronov (Prokhorov General Physics Institute of the Russian Academy of Sciences, Moscow) for valuable discussions. The authors are grateful to the Center for Collective Use of the GPI RAS for the equipment provided.

**Conflicts of Interest:** The authors declare no conflict of interest.

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
