# Peer review of "Cultivation of Solanum lycopersicum under Glass Coated with Nanosized Upconversion Luminophore"

_applsci, doi:10.3390/app112210726_

Round 1
Reviewer 1 Report
The manuscript is interesting and proposes a smart use of up-conversion rare earth nanoparticles.
However, though it is a communication, a few more details should be provided both on the material characterization and on the plant growth experiments. In particular,
- Indexing of the XRD pattern should be provided, along with the SrF2 spatial group.
- It must be mentioned whether the plants were grown from seeds or exposed through coated glass at a specific growth phase.
- Variations of chlorophyll A and B should be reported separately, along with the effects on carotenoids. A bried description of the GreenImage software is adviced.
Finally the journal format for the subsection should be followed and the scale bar of the SEM image should be made better visible
Author Response
Dear Reviewer,
We are very thankful for your comments. We clarified all points in our reply to the reviewer’s comments, and made some additions and corrections in the new version of the manuscript. We hope that the reviewer will find the revised version of our manuscript publishable.
The manuscript is interesting and proposes a smart use of up-conversion rare earth nanoparticles.
However, though it is a communication, a few more details should be provided both on the material characterization and on the plant growth experiments. In particular,
Indexing of the XRD pattern should be provided, along with the SrF2 spatial group.
It must be mentioned whether the plants were grown from seeds or exposed through coated glass at a specific growth phase.
Variations of chlorophyll A and B should be reported separately, along with the effects on carotenoids. A bried description of the GreenImage software is adviced.
Finally the journal format for the subsection should be followed and the scale bar of the SEM image should be made better visible
Below are the answers to your comments.
Indexing of the XRD pattern should be provided, along with the SrF2 spatial group.
We have made changes to the manuscript in accordance with your comments.
It must be mentioned whether the plants were grown from seeds or exposed through coated glass at a specific growth phase.
The plants were grown from seeds. Before the start of the experiment, all plants were grown under the same light conditions. On the seventh leaf stage, both experiment and control plants were placed under glasses coated with fluoroplate polymer with or without photoconversion nanoparticles, respectively. We have made changes to the manuscript in accordance with your comments.
Variations of chlorophyll A and B should be reported separately, along with the effects on carotenoids. A bried description of the GreenImage software is adviced.
Thanks to the Reviewer for the good suggestion to present the data on the content of photosynthetic pigments in plant leaves separately. This data can be a source of valuable information. However, now we cannot perform the corresponding measurements, since the experiment has already been completed, and the plants have been eradicated. On the other hand, it was important for us to measure the relative content of chlorophyll in the leaves without damaging the plants (we used the chlorophyll content meter CL-01 (Hansatech, UK)). This allowed us to avoid the influence of an additional stress factor, which could affect the course of the experiment.
A bried description of the GreenImage software is adviced.
We have made changes to the manuscript in accordance with your comments.
Finally the journal format for the subsection should be followed and the scale bar of the SEM image should be made better visible
We have made changes to the Figure in accordance with your comments.

Reviewer 2 Report
In this submission to Applied Sciences, the authors study the effect of up-converting luminescent nanoparticles coated on the glasses on the productivity of Solanum lycopersicum. The authors find that the cultivation of tomatoes under photoconversion glasses led to increase in plant productivity and acceleration in plant adaptation to ultraviolet radiation. The authors find that plants growing under photoconversion glasses were able to more effectively utilize the absorbed light energy. The authors conclude that the spectral changes induced by photoconversion glasses can accelerate the adaptation of plants to the appearance of ultraviolet radiation.
I find this manuscript to be of interest to researchers in the applied sciences as well as to readers of this journal. As such, I am moderately supportive of publication with a few minor notes. In particular, there has been prior work using nanoparticles and advanced materials to achieve upconversion in various applications, which should be noted:
Nano Research 2012, 5, 770–782
J. Mater. Chem. A 2014, 2, 3389-3398
Specifically, the prior works also used nanoparticles and extended materials to achieve new luminescent properties that can enhance either plants or other light-harvesting systems, which should be noted. With this minor note, I would be receptive towards re-reviewing this again for subsequent publication.
Author Response
Dear Reviewer,
We are very thankful for your comments.
In this submission to Applied Sciences, the authors study the effect of up-converting luminescent nanoparticles coated on the glasses on the productivity of Solanum lycopersicum. The authors find that the cultivation of tomatoes under photoconversion glasses led to increase in plant productivity and acceleration in plant adaptation to ultraviolet radiation. The authors find that plants growing under photoconversion glasses were able to more effectively utilize the absorbed light energy. The authors conclude that the spectral changes induced by photoconversion glasses can accelerate the adaptation of plants to the appearance of ultraviolet radiation.
I find this manuscript to be of interest to researchers in the applied sciences as well as to readers of this journal. As such, I am moderately supportive of publication with a few minor notes. In particular, there has been prior work using nanoparticles and advanced materials to achieve upconversion in various applications, which should be noted:
Nano Research 2012, 5, 770–782
J. Mater. Chem. A 2014, 2, 3389-3398
Specifically, the prior works also used nanoparticles and extended materials to achieve new luminescent properties that can enhance either plants or other light-harvesting systems, which should be noted. With this minor note, I would be receptive towards re-reviewing this again for subsequent publication.
Below are the answers to your comments.
In particular, there has been prior work using nanoparticles and advanced materials to achieve upconversion in various applications, which should be noted:
Nano Research 2012, 5, 770–782
J. Mater. Chem. A 2014, 2, 3389-3398
Specifically, the prior works also used nanoparticles and extended materials to achieve new luminescent properties that can enhance either plants or other light-harvesting systems, which should be noted.
We have made additions and corrections in the new version of the manuscript in accordance with your comments. We hope that the reviewer will find the revised version of our manuscript publishable.

Round 2
Reviewer 1 Report
The authors mostly addressed the criticism